# Advancing Platelet Research Through Live-Cell Imaging: Challenges, Techniques, and Insights

**DOI:** 10.3390/s25020491

**Published:** 2025-01-16

**Authors:** Yuping Yolanda Tan, Jinghan Liu, Qian Peter Su

**Affiliations:** 1School of Biomedical Engineering, University of Technology Sydney, Sydney, NSW 2007, Australia; ytan5599@uni.sydney.edu.au (Y.Y.T.); jinghan.liu-2@student.uts.edu.au (J.L.); 2Heart Research Institute, Newtown, NSW 2042, Australia; 3Charles Perkins Centre, The University of Sydney, Camperdown, NSW 2006, Australia

**Keywords:** platelets, platelet adhesion, platelet activation, platelet aggregation, platelet diseases, live-cell imaging, real-time imaging, macroscale imaging, nanoscale imaging

## Abstract

Platelet cells are essential to maintain haemostasis and play a critical role in thrombosis. They swiftly respond to vascular injury by adhering to damaged vessel surfaces, activating signalling pathways, and aggregating with each other to control bleeding. This dynamic process of platelet activation is intricately coordinated, spanning from membrane receptor maturation to intracellular interactions to whole-cell responses. Live-cell imaging has become an invaluable tool for dissecting these complexes. Despite its benefits, live imaging of platelets presents significant technical challenges. This review addresses these challenges, identifying key areas in need of further development and proposing possible solutions. We also focus on the dynamic processes of platelet adhesion, activation, and aggregation in haemostasis and thrombosis, applying imaging capacities from the microscale to the nanoscale. By exploring various live imaging techniques, we demonstrate how these approaches offer crucial insights into platelet biology and deepen our understanding of these three core events. In conclusion, this review provides an overview of the imaging methods currently available for studying platelet dynamics, guiding researchers in selecting suitable techniques for specific studies. By advancing our knowledge of platelet behaviour, these imaging methods contribute to research on haemostasis, thrombosis, and platelet-related diseases, ultimately aiming to improve clinical outcomes.

## 1. Introduction

The study of live-cell imaging has significantly evolved over recent decades, addressing the limitations of traditional methods that rely on imaging fixed cells. While fixed-cell imaging can provide static snapshots of cell structures, it fails to capture intracellular dynamic changes and intercellular interactions, leaving key aspects of cell physiological mechanisms unobserved. Live-cell imaging compensates for these drawbacks by enabling real-time visualisation of dynamic cellular and subcellular changes, allowing researchers to observe transient events and gain temporal information about complex cell dynamics and underlying mechanisms.

In the field of platelet biology, live-cell imaging has enabled several critical breakthroughs in visualising platelet dynamic processes, such as platelet activation, adhesion, and aggregation, which were previously hard to capture using fixed-platelet techniques. For example, morphological changes in platelets occurring within milliseconds of activation; cytoskeletal reorganisations, such as the formation of dynamic actin nodules during early spreading; signalling transduction, such as the spatiotemporal dynamics of granule release patterns and receptor aggregation during adhesion; and interactions with other blood cells and blood vessels can be observed by using live-platelet imaging [1,2,3]. The discoveries substantially enhance our understanding of the haemostasis and thrombosis processes. Both in vitro and in vivo live-platelet studies revealed that platelets can form a plug at the injured site of the vessel to restore blood flow. However, the same processes can lead to thrombus formation under pathological conditions [4,5,6]. Therefore, real-time imaging is not only a vital tool for observing platelet physiological mechanisms but also holds considerable potential for clinical applications, particularly in diagnosing platelet dysfunction diseases.

Advancements in microscopy and imaging approaches enable researchers to study platelet dynamics further. Platelets were first discovered by Giulio Bizzozero using an optical microscope [7]. Since then, optical microscopy has emerged as a powerful tool in platelet research and the diagnosis of related diseases, gradually revealing the intricate structure and function of platelets [8]. For instance, optical microscopy can diagnose thrombocytopenia by observing abnormal platelet morphology and size [9]. By integrating optical microscopy with microfluidic systems that mimic physiologically relevant environments, platelet adhesion, activation, and aggregation under flow conditions can be monitored [1]. Fluorescence microscopy was then developed and widely utilised to visualise the live-platelet process at the molecular level by targeting specific proteins with fluorophore-conjugated antibodies [9]. However, conventional microscopes are limited by the diffraction of light and lack of sufficient resolution to observe single molecules. With advances in optical physics and the invention of sensitive cameras, nanoimaging techniques emerged as powerful tools to provide information on the dynamics of subcellular structures [10]. Despite the considerable progress in platelet imaging techniques, there is no single microscope that can fully capture all aspects of platelet behaviours. Each method has its strengths and limitations, and it is important to understand which techniques are best suited to observe specific platelet dynamics.

This review explores the challenges associated with live-platelet imaging and offers a comprehensive overview of imaging techniques, from conventional to advanced approaches. By examining the spatiotemporal resolution, applications, and limitations of these techniques, we aim to highlight their potential advancements and the valuable insights they could offer to enhance platelet research and diagnosis of platelet disorders in the clinical sector.

## 2. Challenges in Live-Platelet Imaging

Live-cell imaging has significantly advanced our understanding of platelet function by observing the dynamic behaviours and intracellular processes during platelet adhesion, activation, and aggregation. In real-time live-platelet imaging, selecting an appropriate imaging technique is crucial for obtaining valid and reliable conclusions. Several factors have to be considered, including the spatial resolution to visualise target structures, the imaging speed to capture the transient platelet behaviours, and the viability of platelets throughout the imaging process. Each of these factors influences the overall quality of the result images, and optimising one factor may require compromises in others [11,12].

When focusing on live-platelet imaging, these challenges were magnified due to their unique biological properties. Firstly, platelets are small; resting human platelets have a diameter of about 2 micrometres (µm) and fully spread to a diameter of about 8 µm when adhering to the fixed ligand substrate [13]. Their size makes it difficult to achieve high-resolution imaging of dynamic intracellular protein distribution without compromising functionality. Additionally, as the physiological role of platelets is to stop bleeding, they are highly sensitive to stimuli and can rapidly activate within milliseconds [14,15]. The fast processes require imaging systems to be high in both spatial and temporal resolution. Platelet fragility further complicates imaging, as even slight mechanical stimulation or uncareful manipulation can inadvertently trigger activation, making meticulous handling essential [14]. Lastly, since platelets are fragments of megakaryocyte cytoplasm, they do not have a nucleus, making traditional genetic manipulation impractical [14,16]. As a result, platelets may rely on fluorophores to indirectly target their specific components to be followed during platelet activation [14]. However, fluorophore-conjugated cells introduce the aforementioned complications of phototoxicity and photobleaching, which can severely impact cell viability. Other approaches include genetic manipulation of mouse models to express platelet-specific fluorescence protein or transfection megakaryocytes to differentiate platelets with engineered proteins, but these processes are challenging in terms of time and cost [17]. There are also several new microscopy techniques for label-free platelet imaging, but they have not yet reached the resolution and specificity compared to fluorescent microscopies [14].

Another major concern in live-cell imaging is maintaining cellular health, as many experiments aim to reconstitute the physiological behaviour of cells in vitro. Unfortunately, cellular health has been largely ignored during live-cell imaging. Some degrees of damage to the cell samples are inevitable, particularly since cells are sensitive to phototoxicity during prolonged exposure to concentrated light [12]. Phototoxicity can be exacerbated by fluorophores, which can generate reactive free radicals and singlet and triplet forms of oxygen, further damaging the cells [11,18]. This is especially problematic for tracking single-molecule fluorophores in high-resolution imaging, where high photon energy is required to irradiate cells, greatly increasing the risk of phototoxicity [19]. Another significant challenge to cellular health is photobleaching, as frequent laser excitation to capture the dynamic behaviour of cells may cause fluorescent markers to lose their ability to emit signals over time. Photobleaching can also result in the formation of free radicals and other reactive products [20]. The degree of photobleaching is mainly dependent on the fluorophore that is chosen. The ideal fluorophore must be bright, stable, and well localised while being nontoxic to ensure minimal interference with the sample cell behaviour [12].

In summary, live-platelet imaging presents various challenges. Balancing the requirements for high-resolution imaging, rapid acquisition speed, and cellular health maintenance remain significant obstacles in this field, demanding ongoing innovation in imaging technology.

## 3. Platelet Adhesion, Activation, and Aggregation

This review categorises platelet dynamic behaviours into three stages: adhesion, activation, and aggregation (Figure 1). All three stages are crucial for platelets to form blood clots and maintain haemostasis and vascular integrity. However, under pathological conditions, these processes can contribute to platelet thrombosis, indicating the importance of investigating specific platelet behaviours via live-cell imaging.

Platelet adhesion is the first step of the haemostatic process, initiated by membrane-associated receptors on platelets that engage with exposed extracellular matrix (ECM) proteins at the site of vascular injury [21]. Three key receptors on platelets are involved in adhesion, including glycoprotein (GP) Ib-V-IX, GP VI, and integrin αIIbβ3. The GP Ib-V-IX complex can bind to von Willebrand factor (vWF), enabling platelets to adhere to the vascular wall under high shear, while GPVI and integrin αIIbβ3 can interact directly with collagen and fibrinogen, respectively [22,23]. These adhesive interactions are critical for subsequent activation and aggregation phases that culminate in the formation of a blood clot. Platelets undergo dramatic morphological changes as they adhere to ECM. Initially, platelets change from a discoid to a round shape, then extend their filopodia and lamellipodia before spreading fully into an irregular shape [24]. Numerous studies have been published that suggest a platelet spreading assay is the essential technique for monitoring platelet adhesion and the spreading process on various substrates in either static or flow conditions [25,26]. In combination with live-cell imaging techniques, tremendous progress has been made in visualising the complexity and rapidity of platelet adhesion events.

Platelet activation follows adhesion and is triggered by integrin αIIbβ3 bidirectional signalling. Inside-out signalling induces intracellular signalling cascades, enhances integrin affinities for ligands, and reinforces platelet adhesion and activation [23]. Outside-in signalling is initiated by ligand binding to integrins and triggers intracellular cascades essential for subsequent aggregation [23]. Intracellular molecules undergo dramatic changes triggered by bidirectional signalling, such as cytoskeletal reorganisation, granule secretion, and upregulation of additional receptors on the platelet surface [22]. Real-time live-cell imaging enables the visualisation of spatiotemporal dynamics of these intracellular molecules and signalling in activated platelets, which was previously difficult to capture.

Platelet aggregation is essential for forming stable clots and stopping bleeding. After platelets adhere to the site of damage, they become activated and release signalling molecules that attract and recruit other platelets to the site, facilitating aggregation [27]. Live imaging techniques have been valuable in capturing the dynamics of platelet aggregation, including recruitment of platelets from flow, growth in size and mass of aggregates, and platelet aggregate structures and interactions within platelets. Both in vivo and in vitro studies have contributed to the understanding of platelet aggregation physiologically and pathologically.

Together, the complexity of platelet adhesion, activation, and aggregation highlights the critical role of live-cell imaging in advancing platelet biology.

**Figure 1 sensors-25-00491-f001:**
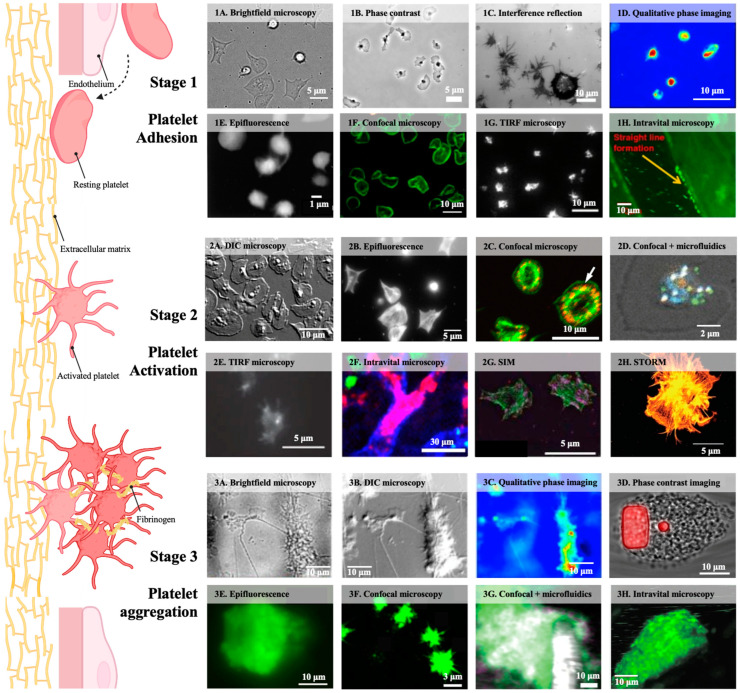
Schematic and representative images of platelet dynamics. Stage 1—platelet adhesion: (**1A**) brightfield [28]; (**1B**) phase contrast [29]; (**1C**) IRM [30]; (**1D**) QPM [31]; (**1E**) epifluorescence [32]; (**1F**) confocal [33]; (**1G**) TIRF [34] showing platelet adhesion and spreading on the substrate; (**1H**) intravital microscopy showing linear platelet adhesion downstream of irradiated vessel area [35]. Stage 2—platelet activation: (**2A**) DIC showing platelets activated by collagen substrate [36]; (**2B**) epifluorescence showing F-actin structures [28]; (**2C**) confocal microscopy showing Pdlim7 (red) and F-actin (green) reorganisation [37]; (**2D**) confocal microscopy with microfluidics showing degranulation labelled by anti-CD63 antibody (green) [38]; (**2E**) TIRF showing actin nodule [39]; (**2F**) intravital microscopy showing activated platelets (red), endothelium (blue), and neutrophils (green) [40]; (**2G**) SIM capturing fixed platelets with F-actin (green), Pdlim7 (magenta), and α-actinin (red) staining [37]; (**2H**) STORM of actin filaments in platelets fixed at 20 min post-activation [41]. Stage 3—platelet aggregation: (**3A**) brightfield [42]; (**3B**) DIC [42]; (**3C**) QPM [42] of the same platelet aggregate formed on collagen; (**3D**) phase contrast [43]; (**3E**) epifluorescence [43] platelets labelled with fluo-3 (green) forming aggregates at the edges of a block (red, DiI); (**3F**) confocal microscopy [44]; (**3G**) confocal microscopy with microfluidics [45]; (**3H**) intravital microscopy [35].

## 4. Techniques for Live-Platelet Imaging

### 4.1. Label-Free Optical Microscopy

Optical microscopy imaging techniques are fundamental tools in platelet biology, providing label-free visualisation of the platelet dynamics when they undergo adhesion, activation, and aggregation. These optical microscopies, including but not limited to brightfield microscopy, differential interference contrast (DIC) microscopy, phase-contrast microscopy, interference reflection microscopy (IRM), and qualitative phase microscopy (QPM), have advanced our understanding of platelet behaviour by capturing real-time events.

#### 4.1.1. Brightfield Microscopy

Brightfield microscopy has the simplest and cheapest setup among these methods, in which light directly illuminates the whole platelet cell body and causes the platelet to appear semi-transparent [46]. It is useful for observing general platelet morphological changes during adhesion [28,47]. To model platelet aggregation in vitro, brightfield microscopy is often combined with ligand-coated microfluidic devices for multiparameter measurements of aggregate formation during perfusion of whole blood or washed platelets at defined wall shear rates (Figure 1(1A)) [28,43,48]. The platelet aggregate formation time, size, and surface area can be quantified and analysed for assessing platelet functions (Figure 1(3A)) [42,48,49]. Additionally, brightfield microscopy is frequently used as a comparative channel for fluorescence microscopy, enabling the investigation of platelet–leukocyte aggregate formation within physiologic microenvironments [50]. However, due to the transparent property of platelets, the acquired images lack contrast and resolution for detailed observation of subcellular structural dynamics during platelet activation.

#### 4.1.2. Phase-Contrast Microscopy

Compared to brightfield microscopy, phase-contrast microscopy has an additional condenser annulus and phase plate that amplifies the contrast of platelets to the background [51]. Phase-contrast microscopy is also frequently used for platelet morphology observations, such as the observation that filopodia extension and platelet spreading depend on the different ligand densities to which they adhere [52]. In addition, the activation and migration of platelets following their involvement in the innate immune response were also revealed by phase-contrast microscopy (Figure 1(1B)) [29]. Phase-contrast microscopy is particularly useful when studying changes in platelet aggregation. Combining phase-contrast microscopy with microfluidics, the contractile force of aggregates under different shear rates and the effects of platelet inhibitors on the force were examined (Figure 1(3D)) [43]. The limitations of phase-contrast microscopy, such as being time-consuming and imprecise, have been indicated [53]. It is not designed for tracking specific molecular interactions, limiting its utility for studies that require insight into protein behaviour or intracellular signalling.

#### 4.1.3. Differential Interference Contrast (DIC) Microscopy

DIC microscopy provides additional contrast and pseudo-3D imaging, giving platelets distinct and shadow-cast appearances [24,54,55]. These pseudo-three-dimensional images can be used to analyse individual platelet characteristics, including spreading area, perimeter, and circularity [41,54]. The ability of DIC microscopy to visualise platelets with enhanced contrast allowed researchers to quantify platelet adhesion alterations associated with storage time or different spreading morphology induced by antiplatelet drugs (Figure 1(2A)) [36,56]. Microfluidic-equipped DIC microscopy revealed that platelets gradually formed aggregates under flow conditions (Figure 1(3B)) [42,57]. Platelet aggregates consist of a high-density core enclosed by a low-density shell, and aggregation kinetics substantially depend on different flow parameters [58]. The technique helps identify the mechanical and physiological formation of platelet aggregates, but it is not suited for molecular-level studies because of limitations in resolution. Unwanted shadow artefacts created by DIC microscopy can also result in challenges in using automated segmentation techniques as they cannot distinguish the real platelet spreading areas by thresholding and edge detection [54].

#### 4.1.4. Interference Reflection Microscopy (IRM)

IRM, also known as reflection interference contrast microscopy, provides detailed information on the contact points between platelets and the substrate through the interference of reflected light waves [30,59,60]. This technique enables spatiotemporal quantitative characterisation of platelet spreading with nanoscale precision and millisecond temporal resolution, which is suitable for studying platelet adhesion dynamics [61]. Mapping and quantification of the dynamic interaction area between platelets and ECM proteins, such as collagen IV and fibrinogen, have been reported in several studies (Figure 1(1C)) [30,59]. Additionally, Reininger et al. used IRM to visualise the interaction between GPIbα with immobilised vWF and their function in mediating the formation of platelet membrane tethers under flow conditions [62]. However, due to the specificity of IRM for observing surface-level interactions, it does not provide information about platelet activation and aggregation.

#### 4.1.5. Quantitative Phase Microscopy (QPM)

QPM collects and quantitates platelet properties such as height and topography by detecting phase shifts as light passes through the cell [51]. It can quantitatively and non-invasively study both morphological and biochemical properties of platelets at the single-cell level [31]. By reconstructing the three-dimensional tomography of individual platelets, changes in mass distribution and thickness during platelet morphological transition before and after treatment of agonists have been shown (Figure 1(1D)) [31]. In addition, the nanoscale sensitivity of QPM to morphology and dynamics enables real-time monitoring of changes in refractive index and mass during aggregation, which makes QPM ideal for studying platelet aggregation under physiological conditions (Figure 1(3C)) [42,63,64]. For example, studies have quantified surface area and mass changes in platelet aggregates under physiological shear rates by QPM [64]. In the clinical context, the study by Klenk et al. used QPM and provided an effective method to assess and predict disease severity in COVID-19 patients through quantitative measurement of platelet aggregation [65]. The main drawbacks of QPM are the complexity of the imaging setup and the lack of molecular specificity [66].

### 4.2. Fluorescence Microscopy

Fluorescence microscopy has the same magnifying properties as optical microscopy while being able to capture the signals emitted by fluorophores. Platelet components can be labelled with various fluorescent probes, and platelet active behaviours can be tracked. In this review, the applications of fluorescence microscopies in platelet biology, including epifluorescence microscopy, confocal microscopy, and total internal reflection fluorescence (TIRF) microscopy, will be discussed.

#### 4.2.1. Epifluorescence Microscopy

Epifluorescence microscopy is a standard tool for monitoring platelet dynamic behaviour. The entire platelet and its contained fluorophores can be illuminated simultaneously by a parallel light beam, generating images with fluorescent contrast [67]. This imaging technique is highly effective for platelet adhesion studies for visualising platelet morphological changes in static and flow conditions (Figure 1(1E)) [32,68,69]. Fluorophores can be attached to various platelet components, such as phospholipid membranes, allowing epifluorescence microscopy to monitor platelet adhesion and spreading on the substrate [69]. In the dynamic platelet activation process, epifluorescence microscopy has been used to capture dramatic calcium oscillations that occur upon firm adhesion to the vWF surface [32]. The reorganisation of the platelet cytoskeleton during spreading has also been visualised by using SiR-conjugated fluorogenic probes, which revealed that with platelet spreading, the F-actin signal becomes increasingly visible and intense (Figure 1(2B)) [28]. Moreover, epifluorescence microscopy provides detailed information on the size and formation rate of platelet aggregates under different wall shear rates (Figure 1(3E)) [43,70,71]. Research has shown that single platelets adhere and accumulate to form unstable and fibrillar-connected aggregates at low wall shear rates, but the aggregates can be disintegrated by elevated fluid rates [72,73]. The main limitation of epifluorescence microscopy is the high level of out-of-focus light, which alleviates its ability to resolve fine molecular details [74]. This makes it less suited for studies requiring high-resolution imaging of intricate platelet signalling events during platelet activation or protein–surface interactions that drive platelet aggregation.

#### 4.2.2. Confocal Microscopy

Confocal microscopy provides a significant improvement by scanning multiple focal planes to remove out-of-focus light from the image [67]. Images can be sequentially reconstituted by scanning the fluorescence intensity at each detection point of the platelet [67]. This makes confocal microscopy an ideal tool for capturing the three-dimensional (3D) structure of platelet adhesion. In platelet adhesion studies, confocal microscopy allows real-time observation of the dynamic redistribution of individual integrin αIIbβ3 and fibrinogen clusters, as well as the formation of balloon-like structures with the procoagulant surfaces (Figure 1(1F)) [33,59,75,76,77]. Confocal microscopy also enables the co-localisation of multiple fluorescently labelled proteins, enabling the assessment of various functional parameters during platelet activation, including platelet morphology, mitochondrial activity, calcium dynamics, and phosphatidylserine exposure (Figure 1(2C)) [33,37,78]. The study used microfluidic coupled confocal microscopy and discovered that activated platelets retain polyphosphate nanoparticles on the surface after degranulation (Figure 1(2D)) [38]. Moreover, confocal microscopy has greatly enhanced our understanding of platelet aggregation by enabling the reconstruction of multiple optical sections. By staining platelets and fibrin and observing in real-time, studies can map the components of platelet aggregates and dynamics, such as two approaching platelet clusters that could be clustered due to contraction (Figure 1(3F)) [44,79,80]. Marcinczyk et al. provide multiparametric insight into platelet activation status in platelet aggregates using various fluorescent antibodies [81]. Similarly, Chen et al. determined the role of integrin intermediate conformations in promoting biomechanical platelet aggregation by staining different conformations of integrin αIIbβ3 in platelet aggregates and visualising under confocal microscopy compatible with microfluidics (Figure 1(3G)) [45]. Although laser scanning confocal microscopy can provide excellent spatial information, the slow acquisition speed and potential photobleaching limit its ability to capture platelet dynamics in frequent laser excitation or over long periods. Therefore, it is necessary to adjust the parameters during the data collection process and pay attention to the effect of long-term exposure on platelet health.

#### 4.2.3. TIRF Microscopy

TIRF microscopy is a near-field imaging technique that selectively excites fluorophores at a shallow angle near the platelet–substrate interface [67]. This selective excitation eliminates out-of-focus light, providing a high-resolution optical sheet with an improved signal-to-noise ratio [67]. TIRF has been used extensively to observe dynamic interactions between platelet receptors and immobilised substrates, such as GPVI clustering on collagen or integrin αIIbβ3 engaging with fibrinogen during platelet adhesion and spreading (Figure 1(1G)) [34,82,83,84]. TIRF can also be coupled with the microfluidic perfusion system to detect platelet adhesion footprints and interactions with the adhesion ligand under flow conditions [85,86]. In addition, intracellular signalling in activated platelets can also be captured by TIRF. Proteins such as WASp and Pdlim7 were revealed to regulate actin cytoskeletal and nodule formation during platelet adhesion and spreading at the substrate (Figure 1(2E)) [37,39]. While the shallow excitation angle of TIRF microscopy allows precise location of signalling events in spatial resolution, such as granule secretion near the basal membrane, the limitation makes this technique less effective for platelet aggregates, whose structure may extend beyond the thin imaging plane [87].

### 4.3. Advanced Microscopy

Advanced imaging techniques, including intravital and super-resolution microscopy, offer novel insights into platelet behaviour in the physical environment or at ultrastructural resolution. These techniques overcome several limitations of conventional optical and fluorescence microscopy, enabling the visualisation of dynamic platelet ultrastructures and enhancing understanding of molecular mechanisms during platelet adhesion, activation, and aggregation.

#### 4.3.1. Intravital Microscopy (IVM)

In contrast to the imaging techniques previously discussed, intravital microscopy (IVM) allows for real-time observation of platelet dynamic behaviour in vivo, providing a true window and direct view of their interplays with other cell types and their native microenvironment [88,89]. The early conventional optical IVM, such as those employing epifluorescence and confocal microscopy, have been widely used to visualise platelet dynamic behaviours in animal models, particularly for arterial and venous thrombosis [90,91,92]. Platelet thrombus formation was first identified by IVM in the 19th century, with observation of dynamic thrombus growth, fragmentation, and reconstitution [93]. By using fluorescence antibodies to label individual platelets, intravital confocal microscopy enables real-time visualisation of platelet tether, translocation, and accumulation on the surface of the developing thrombi [40,90,91]. These studies also reported the function of fibrin and vWF in facilitating platelet aggregation (Figure 1(3H)) [91,92]. However, the conventional IVM has many limitations, including restricted depth of field and high background signals caused by high optical scattering [94,95]. Two-photon intravital microscopy has overcome these limitations by employing longer excitation wavelengths, limiting fluorescence to a single optical plane, therefore minimising background noises and reducing photobleaching and photodamage [96]. Platelet adhesion can be monitored by two-photon microscopy following the induction of vessel injury by laser irritation, revealing temporary translocation of platelets along the vessel wall and linear platelet adhesion formed in the injured area (Figure 1(1H)) [35,96]. Multiphoton microscopy is a further improvement that extends imaging capabilities by using infrared pulse light for multiphoton excitation of fluorophores and integrating second- and third-harmonic generation to visualise platelet thrombus formation at various focal planes [97,98,99].

Despite its advantages, the spatial resolution of IVM is limited compared to other microscopies, making it challenging to resolve single molecules and subcellular structures within activated platelet aggregates. Moreover, platelets are challenging to genetically modify to express fluorescent signals for intravital imaging as they lack nuclei. As a result, imaging in IVM studies typically relies on infusing fluorescent proteins or genetically modified animals, which might impair platelet function or be time-consuming and costly to culture.

#### 4.3.2. Super-Resolution Microscopy (SRM)

Super-resolution microscopy (SRM) has overcome the resolution limits of conventional microscopy, significantly expanding the scope of platelet research and diagnostics. A comprehensive review has summarised various SRM techniques for advancing platelet research and diagnosing platelet-related disorders [100]. This review will highlight several promising SRM techniques for live-platelet imaging, focusing on stimulated emission depletion (STED) microscopy, structured illumination microscopy (SIM), and stochastic optical reconstruction microscopy (STORM).

The principle of STED microscopy is to use a high-intensity doughnut-shaped laser beam superimposed with a focused excitation laser beam, which can effectively shrink the excitation volume and increase the imaging resolution [101,102,103]. STED microscopy has been successfully used in several living cells. For example, time-lapse STED microscopy recorded endoplasmic reticulum (ER) morphological change in the living mammalian cell and vesicle mobility within neuron axons over time [104,105]. There is limited literature applying STED microscopy to living platelets, as STED microscopy requires relatively high laser power and the need to capture two or more images, which may result in phototoxicity that compromises platelet function [103,106]. Nonetheless, STED has provided valuable insights from fixed-platelet studies, which discovered the stored pattern of angiogenesis regulatory proteins in α granules and the clustering of P-selectin in cancer-cell-exposed platelets [106,107].

SIM is the most widely used in live-cell super-resolution imaging, allowing samples, antibodies, and probes prepared for conventional microscopy to be used [108,109]. SIM is an optical sectioning technique in which multiple-direction images are taken with structured light, and accurate images are obtained by computationally reconstructing the acquired images [108,109]. In platelet studies, time-lapse SIM imaging has shown that the actin polymerisation process was disrupted in Pdlim7-deficient platelets (Figure 1(2G)) [37]. In addition, by combining SIM with molecular force microscopy, the study suggested that platelet force generation and alignment are tightly coordinated processes that occur on different time scales during platelet spreading [110]. While SIM has been effectively used for live cells, much of the literature fixed the platelets to visualise their internal organisations. For example, SIM has provided detailed imaging of platelet granule abnormalities in patients with acute ischemic stroke and Hermansky–Pudlak syndrome [108,111]. The study by Xu et al. used SIM and revealed detailed nanoscale distribution patterns of various subcellular structures in platelets, including mitochondria, dense granules, α-granules, and microtubules [112]. These studies indicate that SIM provides new perspectives on both healthy and pathological platelet functions and offers valuable references for related research and disease diagnosis in the future. SIM has drawbacks, including lower spatial resolution than other super-resolution techniques, and often requires brighter fluorophores, higher laser power, or longer exposure times, which significantly increase the risk of photobleaching and phototoxicity in live-cell studies [113].

STORM is a localisation-based nanoscopy technique that precisely localises single fluorophores using sequentially on-and-off probes [101]. Specifically, it achieves high-resolution images by randomly opening a small subset of fluorescent probes in a field, locating the positions of fluorophores, and repeating the process to combine their localisation [101]. Direct STORM (dSTORM) is the most widely used technique due to its simplicity and ability to use the optical sectioning capabilities of TIRF [109,114]. While STORM has been applied extensively to study various organelles from living cells, such as actin cytoskeleton in live HeLa cells and histone protein in live mammalian cells, its use in live-platelet imaging is still limited [115,116]. Most platelet studies used STORM for fixed platelets and captured their dynamic changes by fixing platelets at different spreading stages and time points. These studies have shown the ultrastructural reorganisation of various organelles during platelet shape changes and activation processes (Figure 1(2H)) [41,117]. The spatial organisation and dynamics of platelet subcellular structures during their development are also revealed by STORM using different fixed morphologies of platelet intermediates [11]. STORM has also been applied to study platelet disorders, such as Glanzmann thrombasthenia, and revealed disrupted actin organisation in these abnormal platelets [118]. Several challenges are faced in live-platelet imaging. Firstly, a single reconstruction of the high-resolution images needs multiple individual frames, making it too slow for real-time live-cell imaging [119]. Platelet adhesion, activation, and aggregation are highly dynamic and transient processes, and changes between captures can result in inaccurate reconstructions [101]. Additionally, sample preparation is critical for obtaining high-quality STORM images, and quality antibodies and adequate labelling are required to minimise the background [109].

## 5. Discussion

The imaging techniques summarised in this review provide promising avenues for future studies of platelet biology and the diagnosis of related diseases. Each imaging method has strengths and limitations (Table 1), and no single imaging technique is ideal for all applications. Appropriate techniques should be chosen depending on the particular platelet behaviours to be investigated.

Optical microscopes can provide non-invasive and label-free imaging of platelet processes. These techniques are adequate for basic studies of platelet morphological changes and aggregate formation but lack molecular specificity, which makes them unsuitable for detailed subcellular structural and intracellular signalling studies. Other challenges, such as shadow artefacts or poor contrast due to the transparency of platelet samples, make precise quantification difficult. Recent advancements in integrating machine learning algorithms have enabled platelet results to be analysed automatically. For example, Kempster et al. utilised the convolutional neural network (CNN), which allowed precise and consistent quantification of the platelet spreading area captured by DIC microscopy [54]. Pike et al. discovered an automatic workflow that can robustly segment individual platelets to quantify their spread surface area and circularity [120]. By combining machine learning into the analysis workflow, user bias and time consumption can be minimised. Moreover, the recently developed isotropic DIC microscopy technique enhances the microscopy performance of phase imaging, enabling more detailed platelet adhesion studies in the future [121].

Fluorescence microscopies allow real-time visualisation of molecular events during platelet activation, but they are often limited by photobleaching, phototoxicity, and high background noises generated by the slow imaging speed and repeated laser excitation. Recent advancements in imaging technology are aiming to overcome these challenges. For example, ribbon scanning confocal microscopy improved imaging efficiency significantly and maintained the same spectral and spatial resolution by capturing strips across the cell [122]. Confocal Raman microscopy (CRM) is another promising technique for live-platelet imaging as it is non-invasive and labelling-free [123,124]. CRM provides molecular compositions of cells by using the phenomenon of inelastic light scattering [123,124]. Its successful applications in whole blood and red blood cell studies suggested its potential to be used in the observation of dynamic platelet behaviours [123,124,125]. To further improve fluorescence microscopy, an automatic noise correction algorithm has been proposed to improve camera performance and enable fast and quantitative imaging with low laser excitation [126]. Additionally, studies have shown that cell photosensitivity can be significantly increased at lower irradiation wavelengths [19]. Phototoxicity increases with decreasing wavelengths but is prominently reduced when fluorophores are excited at wavelengths above 600 nm [19]. This finding makes the development of new red-absorbing fluorophores a possible new direction to improve live-cell imaging. Evidence reveals that near-infrared illumination at 721 nm minimises phototoxicity to living cells [127]. Finally, artificial intelligence software tools are being developed to denoise and restore data [128,129]. While these tools can enhance image quality, the primary goal remains to minimise photobleaching and photodamaging rather than restore the photobleached data.

Intravital imaging compensates for the drawbacks of in vitro imaging, such as artificial platelet activation during sample preparation and the absence of naturally occurring inhibitors in vivo. By enabling real-time observation of platelet behaviour in physiological conditions, intravital imaging offers insights into platelet dynamic interactions with other cell types and microenvironments. However, its insufficient resolution still limits the observation of single-cell dynamics and fine subcellular information within cells. A recent study has expanded the capabilities of intravital imaging by integrating it with computational modelling and provides high spatiotemporal resolution in characterising dynamic platelet clot behaviours [130]. This approach allows for a detailed analysis of platelet thrombus formation, clot stability, and their interaction with various blood flows [130]. In addition, a four-dimensional (4D) imaging platform with enhanced spatiotemporal resolution was described to visualise platelet thrombus components in vivo [131]. Specifically, it has revealed the exposure of phosphatidylserine on platelets and endothelial cells during thrombosis for the first time [131]. Moreover, near-infrared fluorescence (NIRF) is an emerging tool in intravital imaging. NIRF can selectively bind to activated platelets with developed near-infrared fluorescence probes and provide high-resolution images with minimal background interference [132]. This technique provides a powerful modality for assessing platelet dynamics in clinical platelet disease diagnosis.

SRM techniques have surpassed the diffraction limit of conventional microscopy, enabling nanoscale exploration of platelet structure and dynamics. However, the subcellular structure of platelets activates rapidly after stimulation, which requires high capture frequencies to record their dynamic changes, while SRM achieves nanoscale resolution by sacrificing the acquisition time. SIM provides the best compromise between spatial and temporal resolution and low phototoxicity for live-platelet imaging among all super-resolution techniques. For instance, the study used the programmable spatial light modulator to control rapid and precise excitation on the sample plane, which significantly accelerated the acquisition speed of SIM [133]. Li et al. also achieved sub-100 nm resolution of SIM in live-cell imaging by utilising the high-numerical-aperture lens with patterned activation, enhancing imaging details without compromising imaging speed [134]. STED microscopy has also improved imaging time while preserving spatial resolution. Multi-beam interference in STED microscopy has shown that the created optical lattice can provide efficient depletion patterns, which allows rapid imaging and thus improves the capacity of STED for fast and precise visualisation [135]. Additionally, the machine learning algorithm can be used to enhance the resolution of input images from super-resolution microscopy, resulting in a four times improvement in resolution [136]. Lastly, notable advancements have been made in label-free super-resolution methods, including structured illumination, transient absorption, infrared absorption, and coherent Raman spectroscopies [137]. These approaches allow live-platelet imaging without the concerns of phototoxicity and prolonged exposure time but involve trade-offs between spatial resolution.

Advances in live-platelet imaging techniques have significant clinical implications for the diagnosis and treatment of platelet-related disorders. By providing real-time visualisation of platelet dynamics, these techniques bridge the gap between experimental research and clinical applications. For example, live-platelet imaging by brightfield and fluorescence microscopy and microfluidic devices can reveal abnormal platelet aggregation in samples from patients with bleeding disorders [138]. Real-time platelet intracellular signalling monitored by confocal microscopy revealed that platelets with Wiskott–Aldrich syndrome (WAS) showed higher resting calcium levels, more frequent calcium spikes, and more continual loss of membrane potential in mitochondria [33]. In thrombosis management, IVM can be used to evaluate the in vivo effects of antithrombotic therapies. M-tirofiban was found to significantly inhibit the aggregation of platelets and other blood cells in a mouse model of venous thrombosis [139]. Similarly, the infusion of selatogrel can perturb platelet thrombus formation and dissolve existing thrombus but can leave small mural platelet aggregates to seal the blood vessel [140].

New technologies, especially novel imaging modalities [141,142], are emerging to offer insights into high-resolution and functional imaging capacity. Random illumination microscopy (RIM) has high spatial resolution and low phototoxicity comparable to SIM, with lower costs and greater ease of use [143]. This technique is based on speckle illumination and computational image reconstruction, supporting multicolour live-cell imaging for a long period [143]. In addition, DNA-PAINT and tPAINT are potent variants of single-molecule fluorescence microscopy, which emerged as a novel super-resolution technique for live-platelet visualisation. Specifically, tPAINT integrates molecular tension probes with DNA-PAINT, enabling map dynamic mechanical force of integrin αIIbβ3 during platelet spreading [144]. Integrins form a tension ring at the lamellipodia edge and are closely associated with actin cytoskeleton dynamics [144]. tPAINT allows for the acquisition of cell dynamic information and is resistant to photobleaching, offering a promising method for long-term live-platelet imaging. Moreover, combining conventional imaging techniques with these advanced methods may help overcome the challenges of live-platelet imaging, but factors such as cost, technical complexity, and accessibility must be considered as these technologies develop further [145]. Despite the advantages of the imaging techniques mentioned above, electron microscopy (EM) remains a valuable tool for platelet research. They have been used to visualise the arrangement of platelet granules, mitochondria, and actin cytoskeleton with very high precision [146,147]. While EM is unable to perform live-cell imaging due to fixation and dehydration during the sample preparation process, it can be used to complement and validate information from live-cell imaging.

Overall, imaging techniques discussed in this review have provided insight into platelet function at the subcellular and cellular levels by detecting dynamic processes such as platelet adhesion, activation, and aggregation. Although each technique has relative drawbacks, machine learning, novel imaging probes, new labelling methods, and quantitative analysis tools will continue to improve imaging spatiotemporal resolution and specificity. The commercialisation of these advanced approaches will enhance accessibility and enable more precise live-platelet imaging in the future. Together, studying platelet dynamics in real time enhances our understanding of platelet biology and contributes significantly to the understanding of haemostasis, thrombosis, and related pathologies.

**Table 1 sensors-25-00491-t001:** Strengths and weaknesses of live-platelet imaging techniques.

Techniques	Resolution/ Speed	Strengths	Weaknesses	Key Visualisation and/or Findings	Ref.
Optical label-free imaging					
Brightfield microscopy	R: ★☆☆ S: ★★★	- easy to operate - high speed	- low contrast - not suitable for subcellular events	- morphological change under static and flow conditions - aggregate formation	[28,42,43,47,48,49]
Phase-contrast microscopy	R: ★☆☆ S: ★★★	- high speed - high contrast	- no molecular specificity - shadow artefacts - not suitable for subcellular structures	- morphological change and migration on ligand substate - contractile force	[29,43,52,55]
DIC microscopy	R: ★☆☆ S: ★★★	- high speed - pseudo-3D effect	- same as phase-contrast microscopy	- morphological change under static and flow conditions - aggregation kinetics	[36,56,57,58]
Interference reflection microscopy	R: ★☆☆ S: ★★★	- quantitative mapping	- not suitable for platelet activation and aggregation	- quantitative analysis of adhesion dynamics, including contact point and membrane tether	[30,59,60,62]
Quantitative phase microscopy	R: ★☆☆ S: ★★★	- 3D topography - quantitative analysis of morphological properties	- complex imaging setup - no molecular specificity	- single platelet mass, thickness, and surface area changes - mass changes in platelet aggregation for diagnosis of disease severity in COVID-19 patients	[31,63,64,65]
Fluorescence microscopy					
Epifluorescence microscopy	R: ★☆☆ S: ★★★	- high speed - multicolour imaging	- high out-of-focus signal - not suitable for intracellular events	- cytoskeletal reorganisation and calcium signalling during adhesion and activation - aggregate size and formation rates	[32,68,69,70,71,72,73]
Confocal microscopy	R: ★★☆ S: ★★★	- multicolour 3D imaging - no out-of-focus background	- slow acquisition speed - photobleaching and phototoxicity	- dynamics of subcellular structures and activity, calcium dynamics, and phosphatidylserine exposure - aggregate contractions and activation status	[33,38,44,75,76,77,78,79,80]
TIRF microscopy	R: ★★☆ S: ★★★	- high speed - no background	- shallow excitation depth	- dynamic receptor interactions and signalling events near platelet and substrate interface	[37,39,82,83,84,85,86]
Advanced microscopy					
Intravital microscopy	R: ★★☆ S: ★★☆	- in vivo observations - visualise platelet interactions with other cells and the microenvironment	- complexity of the imaging setup - not suitable for intracellular events - challenging in fluorescent labelling	- adhesion, activation, and aggregation in live animals - accumulation on thrombi - interactions with tissue factors, fibrin, vWF, etc.	[35,90,91,92,96,139,140]
Super-resolution microscopy	R: ★★★ S: ★★☆	- high resolution - able to localise single molecules, providing 3D imaging	- low temporal resolution - complexity of the imaging setup - photobleaching and phototoxicity	- ultrastructural distribution of platelet organelles, such as cytoskeleton reorganisation and granule protein expression during platelet spreading	[106,107,110,111,112]

A non-exhaustive table of imaging techniques that can be used for live-platelet imaging. Their resolution and imaging speed are categorised into three levels. For resolution, one star (★☆☆) indicates a resolution limited by diffraction limits, >200 nm, two stars (★★☆) indicate a resolution between 100 and 200 nm, and three stars (★★★) indicate a super-resolution < 100 nm. For imaging speed, one star (★☆☆) represents slow imaging speed (≤1 Hz), two stars (★★☆) represent medium imaging speed (1–10 Hz), and three stars (★★★) represent high-speed imaging at video frequency (>10 Hz).

## Data Availability

Not applicable.

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
