# Peer review of "Advancing Platelet Research Through Live-Cell Imaging: Challenges, Techniques, and Insights"

_sensors, 2025, doi:10.3390/s25020491_

Round 1
Reviewer 1 Report
Comments and Suggestions for Authors
This review summarizes the current platelet imaging techniques, with comprehensive references, and is recommended for publication.
I have one suggestion: could Table 1 specify the exact ranges for imaging speed and resolution for each technique? For example, it would be helpful to include concrete parameters such as "resolution: 1-10 micrometers".
Author Response
Reviewer #1
This review summarizes the current platelet imaging techniques, with comprehensive references, and is recommended for publication.
I have one suggestion: could Table 1 specify the exact ranges for imaging speed and resolution for each technique? For example, it would be helpful to include concrete parameters such as "resolution: 1-10 micrometers".
Response 1: Thanks for the valuable suggestion. We understand the importance of providing concrete resolution ranges, which could help readers better compare the capabilities and applicability of the listed imaging techniques. We have now revised Table 1 to include a new column (the 2nd column in the revised version) specifying the imaging resolution and imaging speed for each imaging technique. We hope this addresses the concern effectively.

Reviewer 2 Report
Comments and Suggestions for Authors
This review manuscript covers various optical microscopy techniques for cell observation and provides useful information to readers. However, as a review article, there is some information that is lacking and needs to be revised.
1. Two-photon microscopes are also used for cell observation and analysis, so it would be better to add a description of this.
2. Recently, live cells had also been observed using confocal laser Raman microscopes, and this content should also be added to the discussion section.
3. The font used for the scale bars in some figures is small, making the numbers unreadable. If possible, please make the font larger.
Author Response
Reviewer #2
This review manuscript covers various optical microscopy techniques for cell observation and provides useful information to readers. However, as a review article, there is some information that is lacking and needs to be revised.
- Two-photon microscopes are also used for cell observation and analysis, so it would be better to add a description of this.
Response 1: Thanks for the insightful comment. We have now included a detailed introduction of two-photon microscopy in the revised main text. This section highlights its working principles, advantages over conventional microscopy, and its applications to monitor platelet adhesion when combined with intravital microscopy (IVM).
Please refer to the revised section:
“Two-photon intravital microscopy has overcome these limitations by employing longer excitation wavelengths, limiting fluorescence to a single optical plane, therefore minimising background noises and reducing photobleaching and photodamage. Platelet adhesion can be monitored by two-photon microscopy following the induction of vessel injury by laser irritation, revealing temporary translocation of platelets along the vessel wall and linear platelet adhesion formed in the injured area (Figure 1.1H).”
- Recently, live cells had also been observed using confocal laser Raman microscopes, and this content should also be added to the discussion section.
Response 2: Thanks for pointing this out. We have now incorporated a discussion of confocal Raman microscopy (CRM) in the manuscript and introduced its potential for future live platelet imaging.
Please refer to the updated discussion section:
“Confocal Raman microscopy (CRM) is another promising technique for live platelet imaging as it is non-invasive and labelling-free. CRM provides molecular compositions of cells by using the phenomenon of inelastic light scattering. Its successful applications in whole blood and red blood cell studies suggested its potential to be used in the observation of dynamic platelet behaviours.”
- The font used for the scale bars in some figures is small, making the numbers unreadable. If possible, please make the font larger.
Response 3: Thanks for pointing this out. We have revised Figure 1 to ensure the font size and scale bars are larger and more readable. We believe this adjustment enhances the clarity of the figures and makes the key details more accessible to the readers.

Reviewer 3 Report
Comments and Suggestions for Authors
The manuscript "Advancing Platelet Research Through Live Cell Imaging: Challenges, Techniques, and Insights" provides a comprehensive and well-structured review of live cell imaging techniques for platelet research. It thoroughly addresses key challenges, explores existing and emerging imaging technologies, and offers valuable insights into their applications in platelet biology. The topic is relevant and timely, especially considering the increasing use of live-cell imaging in biomedical research. However, while the manuscript is informative, there are areas where the content could be further refined for clarity, impact, and accessibility to the target audience. Below are specific comments and suggestions.
1. Some points need to be emphasized, such as why traditional methods are insufficient and what breakthroughs have live-cell imaging enabled in the field.
2. The discussion of phototoxicity and photobleaching is relevant, but a brief mention of potential mitigation strategies (use of red-absorbing fluorophores or low-intensity lasers) would provide practical value to the reader.
3. consider adding a discussion on the clinical implications of live platelet imaging. For example, how could advances in imaging techniques directly impact the diagnosis and treatment of platelet-related disorders?
Author Response
Reviewer #3
The manuscript "Advancing Platelet Research Through Live Cell Imaging: Challenges, Techniques, and Insights" provides a comprehensive and well-structured review of live cell imaging techniques for platelet research. It thoroughly addresses key challenges, explores existing and emerging imaging technologies, and offers valuable insights into their applications in platelet biology. The topic is relevant and timely, especially considering the increasing use of live-cell imaging in biomedical research. However, while the manuscript is informative, there are areas where the content could be further refined for clarity, impact, and accessibility to the target audience. Below are specific comments and suggestions.
- Some points need to be emphasized, such as why traditional methods are insufficient and what breakthroughs have live-cell imaging enabled in the field.
Response 1: Thanks, we agree that this aspect needs stronger emphasis. We have modified the Section 1 (Introduction) in the revised manuscript:
“The study of live cell imaging has significantly evolved over the decades, addressing the limitations of traditional methods that rely on imaging fixed cells. While fixed cell imaging can provide static snapshots of cell structures, it fails to capture intracellular dynamic changes and intercellular interactions, leaving key aspects of cell physiological mechanisms unobserved. Live cell imaging compensates for these drawbacks by enabling real-time visualisation of dynamic cellular and subcellular changes, allowing researchers to observe transient events and gain temporal information about complex cell dynamics and underlying mechanisms.
In the field of platelet biology, live-cell imaging has enabled several critical breakthroughs in visualising platelet dynamic processes, such as platelet activation, adhesion and aggregation, which were previously hard to capture using fixed-platelet techniques. For example, morphological changes of platelets occurred within milliseconds of activation, cytoskeletal reorganisations such as the formation of dynamic actin nodules during early spreading, signalling transduction such as the spatiotemporal dynamics of granule release patterns and receptor aggregation during adhesion, and interactions with other blood cells and blood vessels can be observed by using live platelet imaging.”
- The discussion of phototoxicity and photobleaching is relevant, but a brief mention of potential mitigation strategies (use of red-absorbing fluorophores or low-intensity lasers) would provide practical value to the reader.
Response 2: Thanks for making an excellent point. We decided to edit and modify the discussion section about fluorescence microscopies by the following paragraph:
“Additionally, studies have shown that cell photosensitivity can be significantly increased at lower irradiation wavelengths. Phototoxicity increases with decreasing wavelengths but is prominently reduced when fluorophores are excited at wavelengths above 600 nm. This finding makes the development of new red-absorbing fluorophores a possible new direction to improve live cell imaging. There is evidence revealing that near-infrared illumination at 721nm minimises phototoxicity to living cells.”
- consider adding a discussion on the clinical implications of live platelet imaging. For example, how could advances in imaging techniques directly impact the diagnosis and treatment of platelet-related disorders?
Response 3: This is indeed a crucial aspect that deserves more attention. We have added a new paragraph in Section 5 (Discussion).
“Advances in live platelet imaging techniques have significant clinical implications for the diagnosis and treatment of platelet-related disorders. By providing real-time visualisation of platelet dynamics, these techniques bridge the gap between experimental research and clinical applications. For example, live platelet imaging by brightfield and fluorescence microscopy and microfluidic devices can reveal abnormal platelet aggregation in samples from patients with bleeding disorders. Real-time platelet intra-cellular signalling monitored by confocal microscopy revealed that platelets with Wiskott-Aldrich syndrome (WAS) showed higher resting calcium levels, more frequent calcium spikes and more continual loss of membrane potential in mitochondria. In thrombosis management, IVM can be used to evaluate the in vivo effects of antithrombotic therapies. M-tirofiban was found to significantly inhibit the aggregation of platelets and other blood cells in a mouse model of venous thrombosis. Similarly, the infusion of selatogrel can perturb platelet thrombus formation and dissolve existing thrombus but can leave small mural platelet aggregates to seal the blood vessel.”
